# Discovering a New Okadaic Acid Derivative, a Potent HIV Latency Reversing Agent from *Prorocentrum lima* PL11: Isolation, Structural Modification, and Mechanistic Study

**DOI:** 10.3390/md21030158

**Published:** 2023-02-27

**Authors:** Dong Huang, Lian-Shuai Ding, Fang-Yu Yuan, Shu-Qi Wu, Han-Zhuang Weng, Xiao-Qing Tian, Gui-Hua Tang, Cheng-Qi Fan, Xiang Gao, Sheng Yin

**Affiliations:** 1Southern Marine Science and Engineering Guangdong Laboratory (Zhuhai), School of Pharmaceutical Sciences, Sun Yat-sen University, Guangzhou 510006, China; 2State Key Laboratory of Cellular Stress Biology, School of Pharmaceutical Sciences, Xiamen University, Xiamen 361102, China; 3School of Life Sciences, Xiamen University, Xiamen 361102, China; 4East China Sea Fisheries Research Institute, Chinese Academy of Fishery Sciences, Shanghai 200090, China

**Keywords:** marine toxins, okadaic acid, *Prorocentrum lima* PL11, structural modification, HIV latency reversal activity

## Abstract

Marine toxins (MTs) are a group of structurally complex natural products with unique toxicological and pharmacological activities. In the present study, two common shellfish toxins, okadaic acid (OA) (**1**) and OA methyl ester (**2**), were isolated from the cultured microalgae strain *Prorocentrum lima* PL11. OA can significantly activate the latent HIV but has severe toxicity. To obtain more tolerable and potent latency reversing agents (LRAs), we conducted the structural modification of OA by esterification, yielding one known compound (**3**) and four new derivatives (**4**–**7**). Flow cytometry-based HIV latency reversal activity screening showed that compound **7** possessed a stronger activity (EC_50_ = 46 ± 13.5 nM) but was less cytotoxic than OA. The preliminary structure–activity relationships (SARs) indicated that the carboxyl group in OA was essential for activity, while the esterification of carboxyl or free hydroxyls were beneficial for reducing cytotoxicity. A mechanistic study revealed that compound **7** promotes the dissociation of P-TEFb from the 7SK snRNP complex to reactivate latent HIV-1. Our study provides significant clues for OA-based HIV LRA discovery.

## 1. Introduction

Marine toxins (MTs) are a group of structurally complex and toxicologically diverse secondary metabolites that are mainly produced by algae [1]. While marine toxins pose a risk to human public health through contamination in the food chain, they have also served as drug leads due to potent biological properties [2]. To date, over 100 MTs have been identified and have been structurally divided into three classes: polypeptides, polyethers, and alkaloids [3]. Their intricate structures and significant biological activities have attracted considerable interest from pharmacologists and synthetic chemists. For example, halichondrin B, a complex polyether macrolide originally found in the sponge *Halichondria okadai*, is a potent tubulin assembly inhibitor [4,5,6]. Its fully synthetic macrocyclic ketone analog, eribulin, has been approved by the FDA to treat metastatic breast cancer and liposarcoma. Dolastatin 10, a linear anti-tumor peptide [7,8], once entered up to phase II of clinical trials, but was dropped due to the development of peripheral neuropathy in 40% of patients and insignificant activity in patients with hormone-refractory metastatic adenocarcinoma [9,10]. Nevertheless, its synthetic analogue, brentuximab vedotin, was approved for the treatment of relapsed or refractory Hodgkin lymphoma (HL) and systemic anaplastic large cell lymphoma (ALCL) as an antibody–drug conjugate [11,12]. Palytoxin (PTX), with the most complex natural molecules to date (containing 64 chiral centers), has served as a challenging target in synthetic chemistry for many years [13].

Okadaic acid (OA) is one of the most frequent and worldwide distributed MTs, and it is responsible for diarrheic shellfish poisoning (DSP) in humans [14]. It was originally isolated from the marine sponges *Halichondria okadai* and then was identified as products of dinoflagellates that belong to the genera *Dinophysis* and *Prorocentrum* [15,16]. Structurally, OA is a polyketide compound containing furane- and pyrane-type ether rings and an alpha-hydroxycarboxyl function [15]. At the molecular level, OA is a recognized inhibitor of several types of serine/threonine protein phosphatases (PP), especially with respect to PP type 1 (PP1) and 2A (PP2A) [17]. Thus, it is often used as a useful tool for studying cellular processes that are regulated by the reversible phosphorylation of proteins, including control of the glycogen metabolism, coordination of the cell cycle and gene expression, and maintenance of the cytoskeletal structure. For decades, OA has also been used as a pharmacological tool to elucidate numerous pathways of various diseases (cancer, Alzheimer’s disease, AIDS, etc.) [18,19,20]. However, little progress has been made regarding the medicinal potential of OA due to severe toxic effects, such as cytotoxicity, neurotoxicity, and tumor promoting effect [21].

The existence of latent viral reservoirs is the principal barrier to the eradication of HIV/AIDS. One strategy to overcome this barrier is to use latency reversing agents (LRAs) to reactivate the latent viral properties, which can be eliminated by effective anti-retroviral therapy. So far, although many LRAs have been found to reactivate latent HIV, they have not been clinically used due to high toxicity and poor efficacy. In our ongoing research toward discovering biologically active metabolites from marine sources [22], okadaic acid (OA) (**1**) and OA methyl ester (**2**), two common shellfish toxins, were isolated from the cultured microalgae strain *Prorocentrum lima* PL11. OA could activate the latent HIV, but its severe cytotoxicity restricted its further development as an HIV LRA. Thus, to obtain more tolerable and potent LRAs, we conducted the structural modification of OA by esterification, yielding one known compound (**3**) and four new derivatives (**4**–**7**). HIV latency reversing activities and cytotoxicity screening of OA and these derivates led to the identification of a promising LRA **7**, which possessed a stronger activity (EC_50_ = 46 ± 13.5 nM) but was less cytotoxic than OA. A mechanistic study revealed that compound **7** could efficiently promote the dissociation of the active P-TEFb from the 7SK snRNP complex to reactivate HIV-1 gene transcription elongation. This paper reports the isolation, structural modification, HIV latency reversing activity, and preliminary structure–activity relationship of the HIV latency reversing activity of these compounds, as well as the mechanism of compound **7**.

## 2. Results and Discussion

The microalgae strain of *P. lima* PL11 was cultured, extracted, and partitioned to obtain three fractions. An EtOAc fraction (3.3 g) was purified using various column chromatography and recrystallization methods, obtaining OA (**1**, 150.0 mg) and OA methyl ester (**2**, 4.6 mg) (Figure 1). Then, we conducted the structural modification of OA by esterification, yielding one known compound (**3**) and four new derivatives (**4**–**7**) (Figure 1). The synthetic routes from OA to its derivatives are shown in Figure 1, and the structural modifications occurred at C1-carboxyl and OH-2, 7, 24, and 27 in compound **1**. Briefly, the esterification of C1-carboxyl in **1** using ethanol and benzyl alcohol yielded a known OA ethyl ester (**3**) and benzyl-esterified derivative (**4**), respectively. The acylation of free hydroxyls in compound **1** (OH-2, 7, 24, and 27) via different acyl chlorides or acid anhydride afforded the butyrylated (**5**), benzoylated (**6**), and 2-thiophenecarbonylated derivatives (**7**), respectively. The above structural modification was aimed to investigate the influences of each kind of substituent (alkyls, phenyls, and aromatic heterocycles) at the C1-carboxyl location or free hydroxyls on the cytotoxicity and HIV latency reversing activity of compound **1**. The structures of new compounds were established via analyzing 1D and 2D NMR spectra as well as HRESIMS data, while the known compounds were identified by a comparison of the NMR data with those in the literature.

The HIV latency reversing activity of OA and its derivatives were screened on various model cell lines, including 2D10, NH1, and NH2 cells, which are popular for studying HIV post-integrative latency [23,24]. 2D10 cells are Jurkat-based post-integrative latency model cells, containing a nearly complete HIV genome except for the *nef* that is replaced by the green fluorescent protein (GFP)-coding sequence. Thus, by detecting the expression of GFP in 2D10 cells, we could evaluate the ability of OA derivatives to activate the latent HIV. As shown in Figure 2A, OA exhibited significant HIV latency reversing activity from 15 to 40 nM, but also showed severe cytotoxicity. The cell viability was 20.1% under the treatment of OA at 25 nM (Figure 2B). Following screening, it was indicated that the cytotoxicity of all of these OA derivatives have been remarkably attenuated at 25 nM compared with OA. The cell viability was increased to 74.2% under the treatment of compound **7** at 25 nM. However, their latency reversing activity has also been significantly decreased, except for compound **7**, which exhibited a stronger activity than that of OA, with an EC_50_ of 46 ± 13.5 nM (Figure 2B). Thus, we further assessed the HIV latency reversing activity of compound **7** in NH1 and NH2 cells. NH1 and NH2 are HeLa-based isogenic cell lines, both of which contain an integrated HIV-1 LTR-luciferase reporter gene; however, only the latter stably expresses CMV-Tat-HA. The expression level of luciferase as a reporter gene could reveal the transcription level of the latent HIV-1 gene. NH1 and NH2 cells were treated with different concentrations of compound **7** (NH1: 50, 100, 150, and 200 nM; NH2: 5, 12.5, 25, 50, 75, and 100 nM) for 12 h. The results indicated that compound **7** reactivated latent HIV-1 in a dose-dependent manner in NH1 and NH2 cells (Figure 2C,D), respectively. Furthermore, tat-dependent HIV-1 transcription (NH2) was more sensitive to compound **7** than basal HIV-1 transcription (NH1). The abovementioned results illustrated that compound **7** was an efficient and promising HIV LRA, being more tolerable and potent than OA. 

Structurally, OA is a polyether containing furane and pyrane-type ether rings and an alpha-hydroxycarboxyl function. At the molecular level, OA is a recognized potent PP2A inhibitor and is regarded as a useful tool for biological studies of protein phosphatases [17]. The structure–activity relationship of OA bound to PP2A has been investigated by many groups [25]. For example, substitution at the C1-carboxyl or OH-24 location led to a dramatic decrease in the activity. Hydrogenation at the C14–C15 double bond or deoxidation at C2, which affect the circular conformation, reduces the activity. In the present study, we found that esterification of C1-carboxyl in compound **1** also dramatically decreased the latency reversing efficacy, as exemplified by OA methyl ester (**2**), OA ethyl ester (**3**), and the benzyl-esterified derivative (**4**). All of three compounds could only slightly activate the latent HIV at 10 μM (Appendix A), indicating that C1-carboxyl group was essential for activity. For hydroxyl groups at C-2, C-7, C-24, and C-27, different substituents resulted in a latency reversing efficacy ranking of 2, 7, 24, 27-O-2-thiophenecarbonyl (**7**) > 2, 7, 24, 27-hydroxyl (**1**) > 2, 7, 24, 27-O-benzoyl (**6**) > 2, 7, 24, 27-O-butyryl (**5**) (Figure 2B and Appendix A). These results showed that the C-2-C-7-C-24-C-27 fragment can be hydroxylated or esterified. In addition, analysis of cell viability showed that the esterification of C1-carboxyl (**2**–**4**) or free hydroxyl groups at C-2, C-7, C-24, and C-27 (**5**–**7**) in compound **1** could significantly decrease the cytotoxicity. Briefly, the abovementioned SARs investigation indicated that the C1-carboxyl group in OA was essential for activity, while the esterification of C1-carboxyl or free hydroxyls at C2, 7, 24, 27 was beneficial for reducing cytotoxicity (Figure 2E).

Positive transcription elongation factor b (P-TEFb), which is composed of CDK9 and CycT1, plays a key role in HIV transcriptional activation. While the infected CD4^+^ T cells were in latency, most of the key P-TEFb was sequestered in the inactive 7SK small nuclear ribonucleoprotein (7SK snRNP) complex, which comprises 7SK snRNA, P-TEFb, HEXIM1, LARP7, and MePCE. The release of P-TEFb from this complex is essential for latency reactivation [26]. As **7** exhibited significant HIV-latency-reversing activity, to investigate whether **7** exerted its efficacy by inducing the dissociation of 7SK snRNP complex to release active P-TEFb, we performed an anti-FLAG immunoprecipitation assay in F1C2 cells and examined the components of the 7SK snRNP complex in the immunoprecipitates. As shown in Figure 2F, the total expression of MePCE, LARP7, and HEXIM1 (NE) did not change after treatment with compound **7**. However, compound **7** could significantly reduce the interactions between MePCE, LARP7, or HEXIM1 with CDK9 and could not reduce the interactions between CycT1 and CDK9. In addition, the amounts of super-elongation complex (SEC) subunits AFF4 and ELL2 were improved in NE samples after treatment with compound **7**, but the level of SEC was reduced in the immunoprecipitates. These results indicated that compound **7** could promote the release of active P-TEFb from 7SK snRNP to reactivate HIV-1 gene transcription elongation.

## 3. Materials and Methods

### 3.1. General Experimental Procedures

Optical rotations were measured on a Perkin-Elmer 341 polarimeter (Perkin-Elmer, Waltham, MA, USA). UV spectra were recorded on a Shimadzu UV-2450 spectrophotometer (Shimadzu, Kyoto, Japan). IR spectra were determined on a Bruker Tensor 37 infrared spectrophotometer (Bruker, Bremerhaven, Germany) with KBr disks. NMR spectra were measured on a Bruker AM-400/500 spectrometer (Bruker, Bremerhaven, Germany) at 25 °C. HRESIMS analyses were carried out on a Shimadzu LCMS-IT-TOF spectrometer (Shimadzu, Kyoto, Japan). A Shimadzu LC-20AT (Shimadzu, Kyoto, Japan) equipped with an SPD-M20A PDA detector was used for the HPLC analysis, and a YMC-pack ODS-A column (250 × 10 mm, S-5 μm, 12 nm) (YMC, Kyoto, Japan) or Phenomenex Lux cellulose-2 chiral column (250 × 10 mm, 5 μm, 12 nm) (Phenomenex, Los Angeles, CA, USA) was used for semipreparative HPLC separation. Silica gel (100–200, 300–400 mesh, Qingdao Haiyang Chemical Co. Ltd., Qingdao, China) and Sephadex LH-20 gel (GE Amersham Biosciences, Boston, MA, USA) were used for the column chromatography (CC). All solvents were of analytical grade (Guangzhou Chemical Reagents Company, Ltd., Guangzhou, China) while acetonitrile (MeCN) was of HPLC grade (Grace Chemical Technology Co. Ltd., Qingdao, China).

### 3.2. Microalgae Material

The microalgae strain of *P. lima* PL11 was obtained from Prof. Hong-Long Zhou of National Taiwan University. The voucher specimen is deposited at the East China Sea Fisheries Research Institute, Chinese Academy of Fishery Sciences, P. R. China (Accession number PL11-2014001).

### 3.3. Fermentation and Extraction

The microalgae strain of *P. lima* PL11 was cultured in batches using f/2 medium to obtain a total of 200 L of mixtures at a cell density of 1.5 × 10^8^ cells/L. The mixtures were filtered, and the filter residues (microalgae cells) were extracted with CH_3_OH (15.0 L × 5) and evaporated under reduced pressure to yield a crude intracellular extract. Meanwhile, the filtrates were evaporated and then subjected to HP20 macroporous adsorptive resins eluted with H_2_O and CH_3_OH, successively, to afford the extracellular lipophilic constituents. The intracellular and extracellular extracts were combined, suspended in H_2_O (4 L), and successively partitioned with petroleum ether (PE, 1.5 L × 3), ethyl acetate (EtOAc, 2.0 L × 4), and n-butanol (n-BuOH, 1.5 L × 4), affording the PE fraction (12.1 g), EtOAc fraction (3.3 g), and n-BuOH fraction (6.7 g).

### 3.4. Isolation and Purification

The EtOAc fraction (3.3 g) was applied to a silica gel CC (L 300 × Φ 30 mm, 100–200 mesh) and eluted with CHCl_3_/CH_3_OH in gradient (*v*/*v*, 100:0, 80:1, 60:1, 50:1, 40:1, 30:1, 20:1, 15:1, 10:1, 8:1, each 500 mL) to obtain 10 fractions (Frs. I–X). Fr. VI (0.31 g) was separated by Sephadex LH-20 eluted with ethanol according to the TLC analysis, followed by recrystallization with CH_3_CN to yield the major compound **1** (108.0 mg). Fr. V (0.17 g) was purified by a semi-preparative HPLC method (MeCN/H_2_O, 40:60 → 95:5, YMC-pack ODS-AQ column) to obtain another small amount of compound **1** (42.0 mg, *t*_R_ 13.7 min) and **2** (4.6 mg, *t*_R_ 11.2 min).

### 3.5. Preparation of Compound ***3*** and ***4*** by Esterification of C1-Carboxyl in Compound ***1***

We added a drop of concentrated sulfuric acid to a solution of compound **1** (10 mg, 0.0124 mmol) in freshly anhydrous ethanol (1 mL). The reaction mixture was stirred at rt for 2 h and then quenched by adding excess saturated NaHCO_3_ solution. The resulting solution was washed with H_2_O and then extracted with EtOAc. The concentrated EtOAc residue was purified by semi-preparative HPLC (Phenomenex Lux cellulose-2 chiral column, MeCN/H_2_O = 90:10, 3 mL/min) to obtain the OA ethyl ester product **3** (3.3 mg, *t*_R_ 9.5 min). Compound **3** was a known compound ethylokadaate, which was previously reported [27]. However, its physical constants and spectra data could not be obtained. Herein, we reported the HRESIMS, optical rotation, UV, IR, 1D, and 2D NMR spectra. 

Ethylokadaate (**3**): colorless oil; ${\left[\alpha \right]}_{D}^{20}$ = +18.34 (c 0.3, MeCN); UV (MeCN): λ_max_ (logε) 194 (4.10) nm; IR (KBr) *ν*_max_ 3444, 2925, 2854, 1735, 1454, 1381, 1235, 1175, 1076, 998, 975, 877 cm^−1^; ^1^H NMR (CDCl_3_, 500 MHz) *δ*_H_ 5.52 (1H, dd, *J* = 15.3, 7.6 Hz, H-14), 5.52 (1H, dd, *J* = 15.3, 7.3 Hz, H-15), 5.39 (1H, s, H-41a), 5.33 (1H, s, H-9), 5.06 (1H, brs, H-41b), 4.48 (1H, dd, *J* = 15.0, 7.3 Hz, H-16), 4.12 (1H, brd, *J* = 10.3 Hz, H-24), 4.07 (1H, m, H-27), 3.99 (1H, m, H-4), 3.95 (1H, d, *J* = 10.0 Hz, H-26), 3.66 (1H, m, H-38a), 3.61 (1H, m, overlap, H-12), 3.60 (1H, m, overlap, H-22), 3.55 (1H, m, overlap, H-38b), 3.43 (1H, t, *J* = 10.3 Hz, H-23), 3.39 (1H, m, H-7), 3.29 (1H, dd, *J* = 10.2, 1.5 Hz, H-30), 2.29 (1H, m, H-13), 2.19 (1H, m, H-17a), 2.05 (2H, m, H-3a and 18a), 2.01 (1H, m, H-21a), 1.94 (1H, m, overlap, H-29), 1.93 (2H, m, overlap, H-11a and 20a), 1.89 (1H, m, overlap, H-36a), 1.88 (1H, m, overlap, H-18b), 1.85 (6H, m, overlap, 6a, 6b, 11b, 20b, 21b, and 32a), 1.80 (1H, m, H-31), 1.74 (1H, m, H-5a), 1.73 (3H, s, H-43), 1.69 (1H, m, H-3b), 1.65 (1H, m, H-35a), 1.62 (1H, m, H-37a), 1.60 (1H, m, overlap, H-17b), 1.58 (1H, m, H-33a), 1.54 (1H, m, H-36b), 1.51 (1H, m, 37b), 1.44 (1H, m, H-35b), 1.40 (1H, m, overlap, H-33b), 1.37 (3H, s, H-44), 1.35 (1H, m, overlap, 32b), 1.33 (1H, m, overlap, H-28a), 1.27 (1H, m, H-5b), 1.06 (3H, d, *J* = 7.0 Hz, H-40), 1.03 (3H, d, *J* = 7.0 Hz, H-42), 1.00 (1H, m, overlap, H-28b), 0.93 (3H, d, *J* = 6.5 Hz, H-39), for 1-ethyl ester: 4.34 (1H, m, H-1′a), 4.19 (1H, m, H-1′b), 1.31 (3H, t, *J* = 7.4 Hz, H-2′); ^13^C NMR (CD_3_Cl_3_, 125 MHz) *δ*_C_ 176.6 (C-1), 144.0 (C-25), 138.7 (C-10), 135.7 (C-14), 131.1 (C-15), 121.7 (C-9), 112.5 (C-41), 105.8 (C-19), 96.1 (C-8), 95.6 (C-34), 84.9 (C-26), 79.1 (C-16), 76.6 (C-23), 75.3 (C-2), 75.1 (C-30), 71.6 (C-7), 71.0 (C-12 and 24), 69.6 (C-22), 68.7 (C-4), 64.6 (C-27), 60.3 (C-38), 43.8 (C-3), 41.9 (C-13), 37.2 (C-18), 35.9 (C-35), 35.3 (C-28), 33.0 (C-11 and 20), 31.8 (C-5), 31.1 (C-29), 30.7 (C-17), 30.3 (C-33), 27.5 (C-31), 27.4 (C-44), 27.2 (C-6), 26.5 (C-21), 26.4 (C-32), 25.5 (C-37), 23.0 (C-43), 18.8 (C-36), 16.2 (C-40), 15.9 (C-42), 10.7 (C-39), for 1-ethyl ester: 61.5 (C-1′), 14.2 (C-2′); HRESIMS m/z 855.4833 [M + Na]^+^ (calcd for C_46_H_72_O_13_Na^+^, 855.4865).

We added benzyl alcohol (5.16 μL, 0.0496 mmol), 4-dimethylamino-pyridine (DMAP, 6.04 mg, 0.0496 mmol), and 1-(3-Dimethylaminopropyl)-3-ethylcarbodiimide hydrochloride (EDC, 7.70 mg, 0.0496 mmol) to a solution of compound **1** (20 mg, 0.0248 mmol) in freshly anhydrous dichloromethane (1 mL). The reaction mixture was stirred at rt for 6 h and then evaporated. The residue was dissolved in methanol and purified by silica gel CC (L 200 × Φ 10 mm, 300–400 mesh) eluted with CHCl_3_/CH_3_OH in gradient (*v*/*v*, 100:1, 80:1, 50:1, 40:1, 30:1, each 100 mL) to obtain compound **4** according to the TLC analysis (12.2 mg).

Benzyl-esterified derivative of OA (**4**): colorless oil; ${\left[\alpha \right]}_{D}^{20}$ = + 13.33 (c 0.3, MeCN); UV (MeCN): λ_max_ (logε) 194 (4.30) nm; IR (KBr) *ν*_max_ 3445, 2935, 1735, 1456, 1382, 1235, 1156, 1077, 1045, 998, 975, 916, 877, 736, 698 cm^−1^; ^1^H NMR (CDCl_3_, 400 MHz) *δ*_H_ 5.60 (1H, dd, *J* = 15.3, 8.1 Hz, H-14), 5.50 (1H, dd, *J* = 15.3, 7.6 Hz, H-15), 5.34 (1H, brs, H-41a), 5.32 (1H, s, H-9), 5.02 (1H, brs, H-41b), 4.50 (1H, dd, *J* = 14.9, 7.6 Hz, H-16), 4.12 (1H, brd, *J* = 10.0 Hz, H-24), 4.07 (1H, m, H-27), 3.97 (1H, m, H-4), 3.94 (1H, d, *J* = 10.0 Hz, H-26), 3.66 (2H, m, overlap, H-12 and 38a), 3.59 (1H, m, overlap, H-22), 3.55 (1H, m, overlap, H-38b), 3.44 (1H, t, *J* = 10.0 Hz, H-23), 3.38 (1H, m, H-7), 3.28 (1H, dd, *J* = 10.2, 2.1 Hz, H-30), 2.27 (1H, m, H-13), 2.18 (1H, m, H-17a), 2.04 (1H, m, H-18a), 2.03 (1H, m, H-3a), 2.00 (1H, m, H-21a), 1.94 (1H, m, H-29), 1.88 (1H, m, overlap, H-36a), 1.87 (3H, m, overlap, H-18b, 21b, and 32a), 1.87–1.83 (4H, m, overlap, H-11a, 11b, 20a, and 20b), 1.82 (2H, m, overlap, H-6a and 6b), 1.79 (1H, m, overlap, H-31), 1.73 (1H, m, overlap, H-3b; 3H, s, H-43), 1.68 (1H, m, H-5a), 1.64 (1H, m, H-35a), 1.60 (1H, m, overlap, H-17b), 1.58 (1H, m, H-33a), 1.54 (1H, m, H-36b), 1.52 (2H, m, H-37a and 37b), 1.43 (1H, m, H-35b), 1.40 (1H, m, overlap, H-33b), 1.35 (2H, m, overlap, H-5b and 32b), 1.34 (3H, s, H-44), 1.32 (1H, m, overlap, H-28a), 1.05 (3H, d, *J* = 6.5 Hz, H-40), 0.99 (3H, d, *J* = 6.5 Hz, H-42), 0.98 (1H, m, overlap, H-28b), 0.92 (3H, d, *J* = 6.5 Hz, H-39), for 1-benzyl ester: 7.36–7.33 (5H, m, H-3′–H-7′), 5.28 (1H, d, *J* = 12.3, H-1′a), 5.16 (1H, d, *J* = 12.3, H-1′b); ^13^C NMR (CDCl_3_, 100 MHz) *δ*_C_ 176.2 (C-1), 143.6 (C-25), 138.8 (C-10), 135.5 (C-14), 131.0 (C-15), 121.7 (C-9), 112.5 (C-41), 105.8 (C-19), 96.1 (C-8), 95.6 (C-34), 84.9 (C-26), 79.1 (C-16), 76.6 (C-23), 75.3 (C-2), 75.1 (C-30), 71.6 (C-7), 70.9 (C-12 and 24), 69.6 (C-22), 68.5 (C-4), 64.6 (C-27), 60.3 (C-38), 44.0 (C-3), 41.6 (C-13), 37.1 (C-18), 35.9 (C-35), 35.2 (C-28), 32.8 (C-11 and 20), 31.8 (C-5), 31.1 (C-29), 30.7 (C-17), 30.3 (C-33), 27.4 (C-6 and 31), 27.2 (C-44), 26.5 (C-21), 26.3 (C-32), 25.4 (C-37), 23.0 (C-43), 18.8 (C-36), 16.2 (C-40), 15.8 (C-42), 10.7 (C-39), for 1-benzyl ester: 128.6 × 2 (C-3′ and 7′), 128.4 (C-5′), 128.0 × 2 (C-4′ and 6′), 135.5 (C-2′), 67.0 (C-1′); HRESIMS m/z 917.4998 [M + Na]^+^ (calcd for C_51_H_74_O_13_Na^+^, 917.5022).

### 3.6. Preparation of Compound ***5***, ***6***, and ***7*** by Acylation of OH-2, 7, 24, and 27 in Compound ***1***


We added excess acyl chlorides or acid anhydride to a solution of compound **1** in freshly distilled pyridine (2 mL). The reaction mixture was stirred at rt for 2 h and then quenched by adding 2 mL of H_2_O. After removal of the solvent under vacuum, the residue was purified by a semi-preparative HPLC (Phenomenex Lux cellulose-2 chiral column, MeCN/H_2_O = 90:10, 3 mL/min). The acyl chlorides used in this experiment were butyric anhydride, benzoyl chloride, and 2-thiophenecarbonyl chloride, which were used to obtain the acylated products **5** (3.6 mg, *t*_R_ 12.5 min), **6** (11.5 mg, *t*_R_ 10.4 min), and **7** (4.2 mg, *t*_R_ 11.3 min), respectively.

OH-2,7,24,27-Tetrabutyrylated derivative of OA (**5**): colorless oil; ${\left[\alpha \right]}_{D}^{20}$ = +30.67 (c 0.3, MeCN); UV (MeCN): λ_max_ (logε) 196 (4.84) nm; IR (KBr) *ν*_max_ 2930, 1737, 1459, 1381, 1225, 1181, 1080, 1000, 975, 878 cm^−1^; ^1^H NMR (CDCl_3_, 500 MHz) *δ*_H_ 5.74 (1H, dd, *J* = 15.5, 8.0 Hz, H-14), 5.62 (1H, m, H-27), 5.56 (1H, dd, *J* = 15.5, 7.4 Hz, H-15), 5.41 (1H, brd, *J* = 10.3 Hz, H-24), 5.22 (1H, s, H-9), 5.02 (2H, s, H-41a and 41b), 4.78 (1H, dd, *J* = 11.8, 4.2 Hz, H-7), 4.42 (1H, m, H-16), 4.11 (1H, d, *J* = 9.5 Hz, H-26), 4.10 (1H, m, overlap, H-4), 3.86 (1H, m, H-22), 3.69 (1H, m, H-12), 3.65 (1H, m, H-38a), 3.55 (1H, m, H-38b), 3.55 (1H, t, *J* = 10.3 Hz, H-23), 3.30 (1H, dd, *J* = 9.7, 1.3 Hz, H-30), 2.38 (1H, overlap, H-13), 2.34 (1H, m, H-3a), 2.12 (1H, m, overlap, H-17a), 2.05 (1H, m, H-6a), 2.00 (1H, m, H-32a), 1.96 (1H, m, H-18a), 1.94 (2H, m, overlap, H-11a and 20a), 1.89 (1H, m, overlap, H-3b), 1.87 (1H, m, overlap, H-36a), 1.82–1.79 (4H, m, overlap, H-11b, 18b, 20b, and 31), 1.75 (2H, m, H-5a and 6b), 1.73 (2H, m, H-21a and 32b), 1.68 (3H, s, H-43), 1.63 (1H, m, overlap, H-35a; 3H, s, H-44), 1.59–1.56 (5H, m, overlap, H-5b, 17b, 29, 33a, and 36b), 1.52–1.50 (3H, m, overlap, H-28a, 37a, and 37b), 1.44 (1H, m, overlap, H-35b), 1.40 (1H, m, overlap, H-33b), 1.37 (1H, m, H-21b), 1.10 (3H, d, *J* = 6.5 Hz, H-42), 1.09 (3H, d, *J* = 6.5 Hz, H-40), 1.04 (1H, m, H-28b), 0.88 (3H, d, *J* = 6.5 Hz, H-39), for 2, 7, 24, 27-O-butyryl: 2.39–2.20 (8H, m, overlap, H-2′, H-2″, H-2‴, and H-2′‴), 1.73–1.53 (8H, m, overlap, H-3′, H-3″, H-3‴, and H-3′‴), 1.00–0.88 (12H, t, overlap, H-4′, H-4″, H-4‴, and H-4′‴); ^13^C NMR (CDCl_3_, 125 MHz) *δ*_C_ 172.1 (C-1), 140.5 (C-25), 139.0 (C-10), 133.7 (C-14), 131.4 (C-15), 120.7 (C-9), 112.4 (C-41), 105.9 (C-19), 95.6 (C-8 and 34), 83.3 (C-26), 79.6 (C-2), 79.0 (C-16), 74.5 (C-30), 74.1 (C-23), 72.2 (C-7), 71.5 (C-12), 71.4 (C-24), 70.3 (C-22), 66.8 (C-27), 65.7 (C-4), 60.4 (C-38), 42.9 (C-3), 41.1 (C-13), 36.8 (C-18), 35.9 (C-35), 33.4 (C-28), 32.7 (C-11 and 20), 31.7 (C-5), 31.1 (C-29), 30.5 (C-17), 30.3 (C-33), 27.3 (C-31), 26.3 (C-32), 26.2 (C-21), 25.4 (C-37), 23.9 (C-6), 22.9 (C-43), 22.1 (C-44), 18.2 (C-36), 16.3 (C-40), 16.2 (C-42), 10.6 (C-39). For 2, 7, 24, 27-O-butyryl: 173.2, 173.0, 172.6, 172.6 (C-1′, C-1″, C-1‴, and C-1′‴); 36.5, 36.2, 36.2, 36.2 (C-2′, C-2″, C-2‴, and C-2′‴); 18.7, 18.6, 18.5, 18.5 (C-3′, C-3″, C-3‴, and C-3′‴); 13.7, 13.7, 13.6, 13.5 (C-4′, C-4″, C-4‴, C-4′‴); HRESIMS m/z 1085.6418 [M + H]^+^ (calcd for C_60_H_93_O_17_^+^, 1085.6413). 

OH-2,7,24,27-Tetrabenzoylated derivative of OA (**6**): colorless oil; ${\left[\alpha \right]}_{D}^{20}$ = +31.32 (c 0.3, MeCN); UV (MeCN): λ_max_ (logε) 197 (5.04) nm; IR (KBr) *ν*_max_ 2934, 1719, 1452, 1380, 1315, 1269, 1176, 1112, 1091, 1070, 998, 975, 878, 852, 710 cm^−1^; ^1^H NMR (CDCl_3_, 400 MHz) *δ*_H_ 5.95 (1H, m, H-27), 5.70 (1H, dd, *J* = 15.3, 7.5 Hz, H-14), 5.68 (1H, brd, *J* = 10.3 Hz, H-24), 5.53 (1H, dd, *J* = 15.3, 7.0 Hz, H-15), 5.23 (1H, s, H-9), 5.08 (1H, s, H-41a), 5.06 (1H, s, H-41b), 4.96 (1H, dd, *J* = 11.8, 4.7 Hz, H-7), 4.41 (1H, m, H-16), 4.32 (1H, d, *J* = 9.5 Hz, H-26), 4.26 (1H, m, H-4), 4.08 (1H, m, H-22), 3.81 (1H, t, *J* = 10.3 Hz, H-23), 3.78 (1H, m, overlap, H-12), 3.65 (1H, m, H-38a), 3.55 (1H, m, H-38b), 3.35 (1H, dd, *J* = 10.2, 1.7 Hz, H-30), 2.44 (1H, m, H-3a), 2.37 (1H, m, H-13), 2.16 (1H, m, overlap, H-6a), 2.15 (1H, m, overlap, H-3b), 2.00 (1H, m, H-32a), 1.97 (1H, m, overlap, H-17a), 1.96–1.94 (3H, m, overlap, H-11a, 20a, and 29), 1.93 (1H, m, H-6b), 1.90 (1H, m, H-18a), 1.82 (3H, s, H-44), 1.81 (3H, m, overlap, H-5b, 31, and 36a), 1.73 (3H, m, overlap, H-18b, 21a, and 32b), 1.67 (1H, m, H-28a), 1.63–1.61 (3H, m, overlap, H-5b, 11b, and 20b), 1.55 (3H, s, H-43), 1.54 (1H, m, overlap, H-35a), 1.51–1.49 (4H, m, overlap, H-17b, 33a, 37a and 37b), 1.47 (1H, m, overlap, H-36b), 1.39 (1H, m, overlap, H-35b), 1.35–1.34 (2H, m, overlap, H-21b and 33b), 1.18 (1H, m, overlap, H-28b; 3H, d, *J* = 6.5 Hz, H-40), 1.08 (3H, d, *J* = 6.5 Hz, H-42), 0.87 (3H, d, *J* = 6.5 Hz, H-39), for 2, 7, 24, 27-OBZ: 7.37–8.04 (20H, m, overlap); ^13^C NMR (CDCl_3_, 100 MHz) *δ*_C_ 175.4 (C-1), 140.2 (C-25), 138.9 (C-10), 133.0 (C-14), 131.6 (C-15), 120.7 (C-9), 113.1 (C-41), 106.1 (C-19), 95.7 (C-8), 95.5 (C-34), 83.6 (C-26), 80.4 (C-2), 79.2 (C-16), 74.6 (C-30), 74.4 (C-23), 73.5 (C-7), 72.4 (C-24), 71.3 (C-12), 70.4 (C-22), 67.3 (C-27), 65.7 (C-4), 60.5 (C-38), 43.2 (C-3), 40.9 (C-13), 36.7 (C-18), 35.9 (C-35), 33.7 (C-28), 32.5 (C-11 and 20), 31.7 (C-5), 31.2 (C-29), 30.5 (C-17), 30.3 (C-33), 27.4 (C-31), 26.4 (C-32), 26.2 (C-21), 25.4 (C-37), 24.1 (C-6), 22.9 (C-43), 22.1 (C-44), 18.7 (C-36), 16.4 (C-40), 16.2 (C-42), 10.7 (C-39). For 2, 7, 24, 27-OBZ: 166.0, 166.0, 165.6, 165.3 (C-1′, C-1″, C-1‴, and C-1′‴); 133.2, 133.2, 133.1, 133.0 (C-5′, C-5″, C-5‴, and C-5′‴); 130.5, 130.1, 130.1, 129.8 (C-2′, C-2″, C-2‴, and C-2′‴); 129.8 × 2, 129.8 × 2, 129.6 × 2, 129.5 × 2 (C-3′, C-7′, C-3″, C-7″, C-3‴, C-7‴, C-3′‴, and C-7′‴); 128.5 × 2, 128.4 × 2, 128.3 × 4 (C-4′, C-6′, C-4″, C-6″, C-4‴, C-6‴, C-4′‴, and C-6′‴); HRESIMS m/z 1221.5781 [M + H]^+^ (calcd for C_72_H_85_O_17_^+^, 1221.5787).

OH-2,7,24,27-Tetra-(2-thiophenecarbonylated)-derivative of OA (**7**): colorless oil; ${\left[\alpha \right]}_{D}^{20}$ = +30.25 (c 0.3, MeCN); UV (MeCN): λ_max_ (logε) 270 (4.05), 251 (4.09), 194 (4.10) nm; IR (KBr) *ν*_max_ 2926, 1712, 1525, 1417, 1362, 1259, 1080, 999, 976, 746 cm cm^−1^; ^1^H NMR (CDCl_3_, 500 MHz) *δ*_H_ 5.84 (1H, m, H-27), 5.77 (1H, dd, *J* = 14.8, 7.3 Hz, H-14), 5.57 (1H, d, *J* = 9.0 Hz, H-24), 5.54 (1H, overlap, H-15), 5.23 (1H, s, H-9), 5.10 (1H, s, H-41a), 5.06 (1H, s, H-41b), 4.88 (1H, dd, *J* = 11.0, 4.0 Hz, H-7), 4.47 (1H, m, H-16), 4.29 (1H, d, *J* = 9.0 Hz, H-26), 4.24 (1H, m, H-4), 4.05 (1H, m, H-22), 3.74 (2H, overlap, H-12 and H-23), 3.65 (1H, m, H-38a), 3.54 (1H, m, H-38b), 3.35 (1H, d, *J* = 10.2 Hz, H-30), 2.50 (1H, m, overlap, H-3a), 2.42 (1H, m, overlap, H-13), 2.17 (1H, m, overlap, H-6a), 2.08 (1H, m, overlap, H-3b), 2.05 (1H, m, overlap, H-17a), 2.02 (1H, m, overlap, H-11a), 2.00 (1H, m, H-32a), 1.93 (2H, m, H-6b and 18a), 1.83 (2H, m, overlap, H-11b and 36a), 1.82 (1H, m, overlap, H-31), 1.78 (2H, m, overlap, H-21a and 32b), 1.75 (1H, m, overlap, H-18b; 3H, s, H-44), 1.67–1.65 (5H, m, overlap, H-5a, 5b, 20a, 28a, and H-29), 1.60 (3H, s, H-43), 1.58 (1H, m, overlap, H-35a), 1.53–1.51 (5H, m, overlap, H-17b, 33a, 36b, 37a, and 37b), 1.42 (1H, m, H-35b), 1.39–1.37 (2H, m, overlap, H-21b and 33b), 1.27 (1H, m, H-20b), 1.15 (1H, m, overlap, H-28b; 3H, d, *J* = 6.5 Hz, H-40), 1.10 (3H, d, *J* = 6.5 Hz, H-42), 0.89 (3H, d, *J* = 6.5 Hz, H-39), for 2,7,24,27-O-thiophene-2-carbonyl: 7.70–7.84 (4H, m, overlap, H-3′, H-3″, H-3‴, and H-3′‴), 7.49–7.56 (4H, m, overlap, H-5′, H-5″, H-5‴, and H-5′‴), 7.04–7.10 (4H, m, overlap, H-4′, H-4″, H-4‴, and H-4′‴); ^13^C NMR (CDCl_3_, 125 MHz) *δ*_C_ 176.6 (C-1), 139.9 (C-25), 139.6 (C-10), 133.6 (C-14), 131.4 (C-15), 120.2 (C-9), 113.3 (C-41), 106.0 (C-19), 95.6 (C-8), 95.5 (C-34), 83.4 (C-26), 80.4 (C-2), 79.1 (C-16), 74.5 (C-30), 74.0 (C-23), 73.6 (C-7), 72.6 (C-24), 71.6 (C-12), 70.3 (C-22), 67.8 (C-27), 65.9 (C-4), 60.5 (C-38), 42.9 (C-3), 41.0 (C-13), 36.7 (C-18), 35.8 (C-35), 33.5 (C-28), 32.4 (C-11), 31.9 (C-5), 31.6 (C-20), 31.1 (C-29), 30.3 (C-17 and 33), 27.4 (C-31), 26.2 (C-21 and 32), 25.4 (C-37), 24.0 (C-6), 22.9 (C-43), 22.7 (C-44), 18.7 (C-36), 16.3 (C-40), 16.0 (C-42), 10.6 (C-39). For 2, 7, 24, 27-O-thiophene-2-carbonyl: 161.6, 161.6, 161.3, 161.0 (C-1′, C-1″, C-1‴, and C-1′‴); 133.9, 133.9, 133.6, 133.3 (C-3′, C-3″, C-3‴, and C-3′‴); 133.9–133.0 (C-2′, C-2″, C-2‴, and C-2′‴); 132.8, 132.7, 132.5, 132.5 (C-5′, C-5″, C-5‴, and C-5′‴); 128.0, 127.7, 127.7, 127.6 (C-4′, C-4″, C-4‴, and C-4′‴); HRESIMS m/z 1245.4039 [M + H]^+^ (calcd for C_64_H_77_O_17_S_4_^+^, 1245.4044).

### 3.7. Cell Lines and Antibodies 

All of the cell lines used in this study, including 2D10, NH1, NH2, and F1C2 cells, were either purchased from the American Type Culture Collection (ATCC, Manassas, USA) or were available from the previous work [23]. The antibodies against CDK9, CycT1, HEXIM1, LARP7, MePCE, FLAG, AFF4, and ELL2 have been previously described [28,29].

### 3.8. Flow Cytometry-Based Screening 

The flow cytometry-based HIV latency reversal activity screening was performed as described in our previous reports [30]. The 2D10 cells were seeded into 24-well plates at a density of 2 × 10^5^ cells and were subsequently treated with the indicated concentrations of the test compounds. Cells treated with prostratin or dimeththyl sulfoxide (DMSO) were used as positive or negative controls, respectively. After incubation for 24 h, cells were harvested and washed twice in PBS solution; then, the samples were analyzed by the ATTUNE NXT flow cytometer (Thermo, Waltham, MA, USA) for the viability of 2D10 cell and the percentages of cells expressing GFP.

### 3.9. Luciferase Reporter Assay 

NH1 and NH2 cells grown in the logarithmic phase were seeded into 24-well plates for 12 h and then exposed to dimeththyl sulfoxide (DMSO) or different concentrations of compound **7** for 12 h. Then, cell lysates were prepared and measured using kit E1501 from Promega according to the manufacturer’s instructions, and the numbers of cells were normalized based on their contained α-tubulin levels.

### 3.10. Immunoprecipitation Assay 

The co-immunoprecipitation assay was performed according to our previous description [30]. For anti-FLAG immunoprecipitation (IP), nuclear extracts (NE) were prepared from HeLa-based F1C2 cells treated with compound **7** or DMSO. NE were incubated with anti-FLAG M2 agarose beads (Sigma-Aldrich, St. Louis, MO, USA) overnight before washing and elution. The beads were extensively washed with buffer D 0.15 (20 mM of HEPES-KOH [pH 7.9], 15% glycerol, 0.2 mM of EDTA, 0.2% NP-40, 1 mM of dithiothreitol, 1 mM of PMSF and 0.15 M of KCl) three times and with buffer D 0.1 (20 mM of HEPES-KOH [pH 7.9], 15% glycerol, 0.2 mM of EDTA, 0.2% NP-40, 1 mM of dithiothreitol, 1 mM of PMSF and 0.1 M of KCl) once; the beads were also eluted with 0.1 M of glycine [pH 3.0] and analyzed by WB analysis with the indicated antibodies.

## 4. Conclusions

Over 160 compounds functioning as HIV latency reversing agents (LRAs) have been identified to date, but none of those molecules have been used in clinical practice due to high toxicity and poor efficacy [31]. Therefore, it is still crucial to seek novel and effective LRAs. OA, a marine polyether toxin, is a recognized potent inhibitor of serine/threonine protein phosphatases 1 (PP1) and 2A (PP2A) and is often used as a useful tool for studying various diseases related processes. Although previous research reported that OA could activate latent HIV, the severe toxic effects restricted its further development as an HIV LRA. To discover tolerable and potent LRAs, we performed the isolation of OA from the cultured strain *P*. *lima* PL11 followed by its structural modification, obtaining OA and six derivatives (**2**–**7**). Subsequent flow cytometry-based HIV latency reversal activity screening resulted in the identification of a promising latency reversing agent compound **7**, which possessed a stronger activity (EC_50_ = 46 ± 13.5 nM) but was less cytotoxic than OA. The preliminary SARs indicated that the presence of the C1-carboxyl group in OA was essential to activity, while the esterification of free hydroxyls at C-2, 7, 24, 27 were beneficial for reducing cytotoxicity. The mechanistic study revealed that compound **7** promoted the release of the active P-TEFb from the 7SK snRNP complex to reactivate HIV-1 gene transcription elongation. Summarily, our finding provides significant clues for OA-based HIV LRA discovery.

## Data Availability

Not applicable.

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
