# Peer review of "Discovering a New Okadaic Acid Derivative, a Potent HIV Latency Reversing Agent from Prorocentrum lima PL11: Isolation, Structural Modification, and Mechanistic Study"

_marinedrugs, 2023, doi:10.3390/md21030158_

Round 1
Reviewer 1 Report
The existence of latent viral reservoirs is the principal barrier to the eradication of HIV/AIDS. One strategy to overcome this barrier is to use latency reversing agents (LRAs) to reactivate the latent viral. Thus, searching the LRAs with low toxicity and high efficacy is a research hotspot. Huang, Ding and coworkers found okadaic acid (OA, 1) and OA methyl ester (2) from the cultured microalgae strain Prorocentrum lima PL11 were potential LRAs but withs severe toxicity. Their following structural modification of OA led to five derivatives (3–7). Importantly, 7 possessed a stronger activity but a less cytotoxicity than that of OA. This work is exciting and attracts the attention of researchers all over the world.
I will recommend it to be accepted after minor revision.
Comments:
1. For some parts of the Introduction, please cite references to support your summary, such as the first three sentences and the last one of the first paragraph.
2. Why did the authors want to synthesize the ethyl- and benzyl-esterified and butyrylated, benzoylated, and 2-thiophenecarbonylated derivatives? Why not make any other derivatives such as 2-furanylcarbonyl derivative? Please give an exploration for the purpose, perhaps the inspiration by others’ works.
3. For Scheme 1, it is better to add the yields of these products in the scheme and the equivalents of reagents in the description of ‘Reagents and conditions’.
Others:
1. P2L45: ‘due toxicity’ → ‘due to toxicity’
2. P7L221: Please add the tR values for compounds 1 and 2, and the mass 150.0 mg for compound 1.
Reviewer 2 Report
This paper describes the generation and isolation of okadaic acid derivatives which were tested for latent HIV activity. One such analogue showed good activation of the latent HIV while inducing less cell death than the parent compound.
General comments:
I think it is critical that ‘toxicity’ is used in the right context. Okadaic acid has been shown to be toxic as it causes illness in humans and is very toxic to mice. Is there a proven correlation between cell viability in the assay and toxicity in vivo? If there is, then this should be explained and if not changes are required throughout the paper. It is cell viability or cytotoxicity that is being measured as opposed to true toxicity. Since the target is to develop a pharmacological agent then some comment should be made about the issue of toxicity.
Abstract.
I didn’t think that the abstract could be read in isolation of the paper as it contained specific information. For example hydroxyls at C2, 7, 24 and 27 is not useful without the structure being available. I think it would be more useful to make it less specific along the lines of modification to free hydroxyls generated analogues which …..
In the abstract as well as elsewhere the term detoxification is used. In my mind this refers to a physiological process where a compound is metabolised to generate a less toxic compound. I don’t think it is appropriate in the case of this paper – reduced toxicity or the like would be better.
Introduction
As above information about toxicity should be included. For example on line 45 a compound was dropped due to toxicity – what toxicity was there, how was it tested ? OA has severe toxic effects, what are they ?
Material and methods
From this section it should be possible to recreate the work but in this paper there is not sufficient detail. I think the authors should go back and think what detail is required for someone to pick up the paper and isolate OA and analogues. For example, HPLC detection wavelengths are not given, for column chromatograph the length of columns, volumes of solvents are not given etc.
The NMR data is of good quality. Would it be possible to present this data in a table? By presenting this way the reader could easily see how the modification of structure affected the NMR assignments.
Specific comments:
The title does not read all that well. I think it would be improved by “discovering a new okadaic acid derivative, a potent …..
Line 19: rather than typical shellfish toxins “common shellfish toxins” would be better.
Line 25: “but a less cytotoxicity than that of OA” doesn’t really make sense “but was less cytotoxic than OA” is better I think
Line 25: SARs not defined
Line 27: detoxification I have mentioned above
Line 29: should be “provides a significant clue” or “ provides significant clues”
Line 61: I think lines 61-63 should be rewritten as there were not clear. I think you mean that OA has been used as a pharmacological tool to elucidate ..
Line 80: Structure-activity relationships on what ? rewrite if you mean on HIV latent activity
Lines 84-89: I think most of this information is best suited for the Materials and methods section so you could reduce just to P.lima was cultured, OA and OA methyl ester isolated, structural modification etc.
Line 89: OA on the assay is described later around line 116
Line 119: Since toxicity is such an important part of the story you could add detail (cell viability was increased from ?% for OA to ?% for 7) etc.
Line 142-148: This needs to be rewritten as it is not clear. Something along the lines of Substitution of hydroxyl groups at … resulted in a latency reversing efficacy ranking of … .Analysis of cell viability showed …. These experiments demonstrated that the C1-carboxyl group was essential for ? activity while … reduced cytotoxicity.
Lines 155-174: I really struggled to follow this
Line 384: “or described before” does that mean available from previous work ?
Line 421: “isolation of cultured strain …” I think you mean “isolation of OA from cultured strain .. OA was structurally modified to yield six derivatives ..

Round 2
Reviewer 2 Report
Thank you for addressing my suggestions.
My only remaining issue is with the level of detail in the material and methods section. I am happy with the details given for the semi-preparative HPLC work it was more detail around the column chromatography that I was after.
For example:
Line 218: The EtOAc fraction (3.3 g) was applied to silica gel CC (CHCl3/CH3OH, 100:1-8:1) to give 10 fractions. Fr VI ...
This work cannot be recreated since the length of the silica gel column is not specified and neither are the volumes of solvent and details of the solvent gradient given. While it is stated that the gradient goes from 100:1 to 8:1 how was that constructed? Something like "100 mL of each solvent mixture (100:1, 20:1, 10:1, 8:1) was eluted and ten 50 mL fractions collected" is required. This is true for other column chromatography methods in the paper.
Author Response
Thanks for your kind suggestion, we have added detailed information for silica gel column separation.
Line 217-221: The EtOAc fraction (3.3 g) was applied to a silica gel CC (L 300 × Ø 30 mm, 100‒200 mesh), eluted with CHCl3/CH3OH in gradient (v/v, 100:0, 80:1, 60:1, 50:1, 40:1, 30:1, 20:1, 15:1, 10:1, 8:1, each 500 mL) to give 10 fractions (Frs. I–X). Fr. VI (0.31 g) was separated by Sephadex LH-20 eluted with ethanol according to TLC analysis, and followed by recrystallized with CH3CN to yield the major compound 1 (108.0 mg).
Line 264-266: The residue was dissolved in methanol and purified by silica gel CC (L 200 × Ø 10 mm, 300‒400 mesh) eluted with CHCl3/CH3OH in gradient (v/v, 100:1, 80:1, 50:1, 40:1, 30:1, each 100 mL) to give 4 according to TLC analysis (12.2 mg).